# Multi-Center Benchmarking of a Commercially Available Artificial Intelligence Algorithm for Prostate Imaging Reporting and Data System (PI-RADS) Score Assignment and Lesion Detection in Prostate MRI

**DOI:** 10.3390/cancers17050815

**Published:** 2025-02-26

**Authors:** Benedict Oerther, Hannes Engel, Caroline Wilpert, Andrea Nedelcu, August Sigle, Robert Grimm, Heinrich von Busch, Christopher L. Schlett, Fabian Bamberg, Matthias Benndorf, Judith Herrmann, Konstantin Nikolaou, Bastian Amend, Christian Bolenz, Christopher Kloth, Meinrad Beer, Daniel Vogele

**Affiliations:** 1Department of Diagnostic and Interventional Radiology, Medical Center—University of Freiburg, Faculty of Medicine, University of Freiburg, 79106 Freiburg, Germany; hannes.engel@uniklinik-freiburg.de (H.E.); caroline.wilpert@uniklinik-freiburg.de (C.W.); andrea.nedelcu@uniklinik-freiburg.de (A.N.); christopher.schlett@uniklinik-freiburg.de (C.L.S.); fabian.bamberg@uniklinik-freiburg.de (F.B.); matthias.benndorf@uniklinik-freiburg.de (M.B.); 2Department of Urology, University Hospital of Freiburg, 79106 Freiburg, Germany; august.sigle@uniklinik-freiburg.de; 3Berta-Ottenstein-Programme, Faculty of Medicine, University of Freiburg, 79106 Freiburg, Germany; 4Research & Clinical Translation, Magnetic Resonance, Siemens Healthineers AG, 91052 Erlangen, Germany; robertgrimm@siemens-healthineers.com; 5Digital & Automation Innovation, Siemens Healthineers AG, 91052 Erlangen, Germany; heinrich.von_busch@siemens-healthineers.com; 6Department of Diagnostic and Interventional Radiology, Medical School and University Medical Center OWL, Klinikum Lippe, Bielefeld University, 32756 Detmold, Germany; 7Department of Diagnostic and Interventional Radiology, University Hospital Tuebingen, 72076 Tuebingen, Germany; judith.herrmann@med.uni-tuebingen.de (J.H.); konstantin.nikolaou@med.uni-tuebingen.de (K.N.); 8Department of Urology, University Hospital Tuebingen, 72076 Tuebingen, Germany; bastian.amend@med.uni-tuebingen.de; 9Department of Urology and Pediatric Urology, University Hospital Ulm, 89081 Ulm, Germany; christian.bolenz@uniklinik-ulm.de; 10Department of Diagnostic and Interventional Radiology, University Hospital Ulm, 89081 Ulm, Germany; christopher.kloth@uniklinik-ulm.de (C.K.); sekretariat.radiologie1@uniklinik-ulm.de (M.B.); daniel.vogele@uniklinik-ulm.de (D.V.)

**Keywords:** prostate cancer, artificial intelligence, multiparametric MRI, PI-RADSv2.1, multi-center study

## Abstract

The number of MRI examinations of the prostate currently increases and is expected to rise even further upon the implementation of prostate cancer screening. Reading examinations is time-consuming but could be accelerated by AI algorithms that detect and stratify cancerous lesions according to the PI-RADS classification system. A plethora of algorithms was developed but lacks generalizability, external validation, and robustness when applied in different hospitals and varying MRI scanners. The algorithm tested in this study proved to be accurate and robust in a multi-center setting across different scanners, especially in confidently excluding prostate cancer. This suggests that it could be implemented in human reading to improve efficiency and readers’ confidence.

## 1. Introduction

Multiparametric Magnetic Resonance Imaging (mpMRI) developed into an established tool in the diagnostic pathway of prostate cancer (PCa) due to its implementation in national and international guidelines [1,2,3]. It is recommended in various clinical settings: in case of suspicion for PCa without prior biopsy, in cases with remaining suspicion despite negative biopsy, and before the inclusion of patients in active surveillance. Consequently, the demand for examinations has risen rapidly and sound interpretation of the prostate and potentially cancerous lesions has gained importance [4,5,6]. Computer-aided diagnostic tools may help to compensate for the anticipated increase in examinations and required reading time and offer reassurance in clinical evaluation [7].

Potentially cancerous prostatic lesions detected in mpMRI are classified according to the Prostate Imaging Reporting and Data System lexicon (PI-RADS, currently in version 2.1 [8]). Scores are assigned in accordance with the probability of PCa, whereas 1 represents a very low probability and 5 suggests a very high probability of PCa. Furthermore, clinically non-significant PCa (ncsPCa) is distinguished from clinically significant PCa (csPCa) according to its histopathological composition, as the entities result in different treatment (or management) pathways. Various recent studies suggest a promising performance of AI-based mpMRI assessment for PCa but lack comparability of algorithms or multi-center validation [9]. Furthermore, AI-based imaging assessment bears the potential to reduce inter-rater variability and offers highly standardized reading processes [10].

The aim of this study was to validate the diagnostic accuracy of a commercially available AI algorithm for detection and classification of csPCa in a multi-center setting.

## 2. Materials and Methods

### 2.1. Study Design

The study was conducted retrospectively in three centers of university care in the Department of Radiology in Freiburg, Tübingen, and Ulm (Germany), each blinded to histopathological results. Approval from the local ethics committees was obtained (Ethics committee of Freiburg 20-1256, date of approval: 28 January 2021; Ethics committee of Ulm University 393/21 and 115/22, date of approval: 9 May 2022; Ethics committee of Tuebingen University 437/2021BO2, date of approval: 16 July 2021) and written informed consent was waived due to the retrospective design.

### 2.2. Patients

Representative patients who were referred to the Departments of Radiology due to the clinical suspicion of PCa (positive DRE, PSA-elevation above 4 ng/mL or positive PSA-dynamics of 0.3–0.7 ng/mL/year) between May 2017 and September 2022 were included in this study. In order to ensure a heterogenous collective, a subset from the consecutively acquired examinations at each site was randomly assigned to be analyzed by the algorithm. All patients underwent transrectal ultrasound (TRUS)-fusion guided biopsy (targeted and systematic) for histopathological verification. Exclusion criteria were (a) prior treatment of the prostate regarding prostate cancer, (b) incomplete mpMRI protocol, (c) severe susceptibility or motion artifacts, (d) missing clinical data or histopathological confirmation.

### 2.3. MRI Reading and Protocol

MpMRI of the prostate was conducted and evaluated in clinical routine according to the latest PI-RADS guidelines (Prostate Imaging Reporting and Data System, currently in version 2.1). Studies that were initially analyzed according to earlier PI-RADS versions were reread by a board-certified radiologist. Readers at the sites were blinded for histopathology. All examinations were acquired in 3T scanners (Freiburg: MAGNETOM Vida, Siemens Healthineers, Forchheim, Germany; Tuebingen: MAGNETOM Vida and Prisma fit, Siemens Healthineers, Forchheim, Germany; Ulm: MAGNETOM Skyra and Magnetom Vida fit, Siemens Healthineers, Forchheim, Germany) according to the PI-RADSv2.1 acquisition protocol in supine position [8]. No endorectal coil was used. Prior to the scan, Butylscopolamine was administered intravenously (dose adjustment according to body weight). All patients received intravenous contrast agent with a body weight-dependent dose (Freiburg: 0.2 mL/kg; Gadoteridol; Bracco Imaging, Konstanz, Germany; flow rate 2 mL/s, saline flush 25–30 mL; Tuebingen: 0.1 mmol/kg; Gadovist, Bayer Vital GmbH, Leverkusen, Germany; flow rate 1.5 mL/s, saline flush 20 mL; Ulm: 0.2 mL/kg; Gadovist; Bayer Vital GmbH, Leverkusen, Germany; flow rate 3 mL/s; saline flush 30 mL). T2 weighted axial sequences were utilized for MRI-guided ultrasound fusion biopsy.

### 2.4. Histopathological Verification

Freiburg, Tuebingen:

A trained radiologist segmented the gland and suspicious lesions manually with the supervision of a board-certified radiologist in the UroFusion software environment (Biobot Surgical Ltd., Singapore, V3.0.1). Lesions that were detected in the clinical routine were recorded and reevaluated according to PI-RADSv2.1, if necessary. MRI-guided ultrasound fusion biopsy was conducted in the MonaLisa environment (Biobot Surgical Pte Ltd., Singapore, version 1.0) under laryngeal mask anesthesia. A three-dimensional model of the gland was reconstructed based on the transrectal ultrasound and fused with the segmentation model of the axial T2 sequence by a trained urologist. Targeted biopsies were then taken via transperineal access followed by systematic biopsies according to the Ginsburg scheme [11]. The procedure was previously described in detail [12].

Ulm:

MRI-guided ultrasound fusion biopsy was performed and accomplished with the Artemis system (Innomedicus, Cham, Switzerland) with local anesthesia or under laryngeal mask anesthesia. A three-dimensional model of the gland and the targets was reconstructed based on the transrectal ultrasound and fused with the segmentation model of the axial T2 sequence by a trained urologist. Targeted biopsies were then taken via transperineal access followed by systematic biopsies according to the Ginsburg scheme.

Biopsy samples were analyzed and reported according to the modified Gleason score grading system of the International Society of Urological Pathology (ISUP) [13].

### 2.5. AI Algorithm

Potentially cancerous lesions in the prostate were detected and classified by an AI-reliant research software that is based on biparametric MRI (MR Prostate AI v1.3.3, build date 15 November 2021; Siemens Healthineers, Forchheim, Germany). Both lesions and the prostate gland were automatically segmented and their volume was calculated automatically by the algorithm.

Previous publications described the employed algorithm in detail [14]. In a first pre-processing step, automatic segmentation of the gland in axial T2 weighted and diffusion weighted sequences (DWI) took place prior to image co-registration. Then, a high b-value image (b = 2000 s/mm^2^) and the apparent diffusion coefficient map (ADC) were calculated. Segmentation, co-registered T2 weighted images, high synthetic b-value, and an ADC map were integrated into a 2D image-to-image convolutional neural network (CNN), resulting in an initial heat map of suspicious findings. A second CNN reduced false positive lesions via 3D patch-wise classification [15]. In the last step, the algorithm assigns a preliminary PI-RADS score to the lesions automatically.

### 2.6. Data Collection

After approval from the respective local ethics committee, data was obtained from the original MRI reports regarding overall PI-RADS score, lesion localization of described lesions, and the individual lesions’ PI-RADS categories. Volumes of the prostate gland and target lesions were derived from the segmentations of the biopsy software (see above). From the AI output, we collected the gland and lesion volume, the overall PI-RADS category, localization of described lesions, and the assigned PI-RADS categories on the lesion level. We manually compared the lesion localizations and categories described by the radiologists with the localizations and categories of the AI results (a visual overlap with the manually defined lesions ≥ 25% was considered as positive detection for the AI system). Comparison was performed by a reader who was blinded to the histopathological results. Furthermore, lesions that were detected additionally by the AI were recorded.

### 2.7. Statistics

Statistical analysis was carried out in the R software environment (version 4.4.2) [16]. Diagnostic accuracy was calculated employing the ‘caret’ package and compared using the McNemar’s test [17]. The Shapiro–Wilk test was used to test for normal distribution [18]. The Wilcoxon Signed-Rank test for matched samples was used to test for a difference in paired samples [19].

## 3. Results

### 3.1. Subjects

The final cohort consisted of 91 patients with a total of 138 lesions. Overall median age was 67 years (range: 49–82). Median PSA prior to biopsy was 8.4 ng/mL (range: 1.47–73.7). Mean PSA density derived from the gland volume measured by the radiologist was 0.22 ng/mL^2^ compared to 0.24 ng/mL^2^ calculated by AI. Table 1 provides an overview of the study population.

### 3.2. Lesion-Level Analysis

Of 138 lesions, 56 (41%) showed benign results after the targeted biopsy. Three lesions (2%) resulted in ISUP 1 and 79 lesions (57%) showed ISUP > 1. Considering that the AI algorithm does not assign PI-RADS 1 and PI-RADS 2 scores, the following PI-RADS scores were allocated (radiologist vs. AI): 0 PI-RADS 1 lesions, 14 vs. 0 PI-RADS 2 lesions, 35 vs. 9 PI-RADS 3 lesions, 41 vs. 19 PI-RADS 4 lesions, and 48 vs. 58 PI-RADS 5 lesions. A total of 52 lesions were not detected by AI. Detected target lesions with allocated PI-RADS scores and corresponding ISUP grades are depicted in Figure 1 for radiologist and AI, respectively. This resulted in a sensitivity, specificity, positive predictive value (PPV), and negative predictive value (NPV) of 97%/86%, 20%/70%, 62%/79%, and 86%/79% (radiologists/AI reading) for a threshold of PI-RADS > 2 compared to 90%/81%, 70%/78%, 80%/83%, and 84%/75% for a threshold of PI-RADS > 3 regarding csPCa.

Two PI-RADS 2 lesions reported by the radiologists harbored csPCa, of which one was detected and classified correctly as PI-RADS 4 and one could not be detected by AI. AI on the other hand missed 11 lesions of csPCa, all of which were correctly classified as PI-RADS > 2 by the radiologist. AI detected 0.38 additional lesions on average (range: 0–4; SD: 0.75).

### 3.3. Patient-Level Analysis

Of 91 patients, 24 (26%) showed benign results after targeted and systematic biopsy. Four cases (4%) resulted in ISUP 1 grading and 63 patients (69%) showed ISUP > 1. The following PI-RADS scores were assigned on the patient level (radiologist vs. AI): 0 patients with PI-RADS 1, 9 vs. 0 patients with PI-RADS 2, 14 vs. 5 patients with PI-RADS 3, 27 vs. 16 patients with PI-RADS 4 and 41 vs. 53 patients with a PI-RADS score of 5. A total of 17 suspicious cases detected by radiologists were not picked up by the AI. PCa detection on the patient level with allocated PI-RADS scores and corresponding ISUP grades are depicted in Figure 2 for radiologist and AI, respectively. This resulted in a sensitivity, specificity, positive predictive value (PPV), and negative predictive value (NPV) of 98%/97%, 29%/54%, 76%/82%, and 89%/88% (radiologists/AI reading) for a threshold of PI-RADS > 2 compared to 92%/91%, 64%/57%, 85%/83%, and 78%/73% for a threshold of PI-RADS > 3 regarding csPCa.

One case with an overall score of PI-RADS 2 harbored csPCa that was detected and classified correctly as PI-RADS 4 by AI. AI, on the other hand, missed two cases of csPCa, both of which were correctly classified as PI-RADS > 2 by the radiologist. Table 2 shows details of PI-RADS scoring and diagnostic accuracy on the lesion level and the patient level. An example case of a complete match of lesions detected by AI and the radiologist is shown in Figure 3, and a false positive lesion detected by the AI is shown in Figure 4.

### 3.4. Example Cases

**Figure 3 cancers-17-00815-f003:**
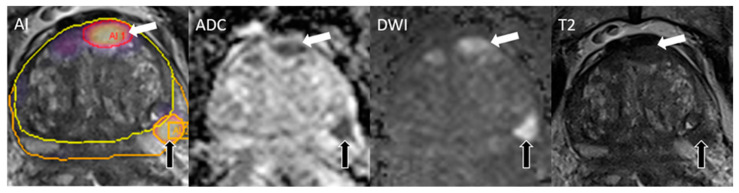
Complete match of lesions detected by AI and radiologist. CsPCa was present in both detected lesions. PSA density at the time of the MRI was 0.15 ng/mL^2^. The lesion in the transitional zone (white arrow) was rated as PI-RADS 5 and the lesion in the peripheral zone (black arrow) was rated as PI-RADS 4 by both the reader and the AI. Both lesions harbored ISUP 2 PCa. Prostate boundaries are segmented with an orange line, TZ with yellow line.

**Figure 4 cancers-17-00815-f004:**
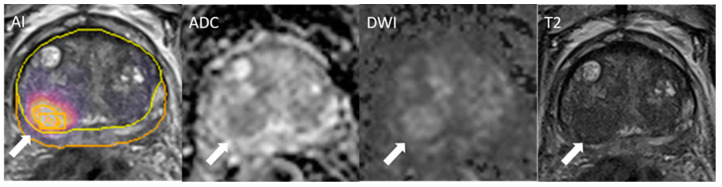
False positive lesion (white arrow) detected by the AI. PI-RADS 4 assigned by AI and PI-RADS 2 assigned by the reader. Histopathology showed benign histopathology. PSA density at the time of the MRI was 0.11 ng/mL.

## 4. Discussion

In this multi-center study, AI-augmented PCa detection proved robust diagnostic accuracy and in parts also better specificity in comparison to human reading. Only two cases of csPCa were missed by the AI on the patient level. These results are in line with current studies also putting the focus on high sensitivity and low false positive rates [20].

Cancer detection rates (CDR) in our study were slightly discrepant to recently published estimates in larger cohorts but showed similar tendencies. Pooled CDR for csPCa in human readings settled at 4% for PI-RADS 2, 20% for PI-RADS 3, 52% for PI-RADS 4 and 89% for PI-RADS 5 on lesion level, and 9% for PI-RADS 2, 16% for PI-RADS 3, 59% for PI-RADS 4, and 85% for PI-RADS 5 on patient level in our previous meta-analysis [21]. In this study, we found 14% for PI-RADS 2, 17% for PI-RADS 3, 61% for PI-RADS 4, and 96% for PI-RADS 5 on lesion level, and 11% for PI-RADS 2, 29% for PI-RADS 3, 67% for PI-RADS 4, and 98% for PI-RADS 5 on patient level for human readers compared to 44% for PI-RADS 3, 68% for PI-RADS 4, and 88% for PI-RADS 5 on lesion level, and 80% for PI-RADS 3, 63% for PI-RADS 4, and 89% for PI-RADS 5 on patient level for AI detection. The highest divergences in CDR were observed in PI-RADS 3 scores on both lesion and patient levels, although implying limited generalizability due to the low number of cases in this category. Furthermore, the modest amount of PI-RADS 3 scores in AI detection speaks against any excessive upgrading from PI-RADS 2. In general, both human readers and AI showed higher CDR compared to current estimates in almost every PI-RADS category on both lesion and patient levels. Interestingly, human readings achieved exceptionally high CDR in PI-RADS 5 categories, whereas the algorithm produced slightly more false positive results which seems to correspond better to current observations.

Sensitivity and specificity for csPCa were 92%/64% for radiologists vs. 91%/57% for AI detection (cut-off PI-RADS ≥ 4) and 98%/29% vs. 97%/54% (cut-off PI-RADS ≥ 3) on patient level. Park et al. reported a comparable pooled sensitivity of 87% and a noticeably higher specificity of 74% (mixed cut-off values) in their meta-analysis on patient level [22]. Their meta-analysis was comparable with mean ages of 50.5–73 years and mean PSA levels of 5.8–13.7 ng/mL compared to a mean of 66.6 years and PSA levels of 10.89 ng/mL in our study. In subgroup analyses for specific cut-offs, lower sensitivity (81%) but higher specificity (82%) was found compared to both human and AI readings for a threshold of PI-RADS ≥ 4. Applying a threshold of PI-RADS ≥ 3 resulted in similar sensitivity (94%) for human and AI readings, and specificity comparable to AI reading (56%), although the human reading in our study showed lower specificity for this threshold. Drost et al. reported both lower sensitivity (91%) and specificity (37%) for csPCa in their meta-analysis, comparing MRI to template biopsies in biopsy-naive and repeat-biopsy settings [23].

AI missed 52 lesions (of which 11 were csPCa: 8% of all lesions), which resulted in 17 missed cancers on patient level (of which 2 csPCa: 2% of all patients; one case of ISUP 3 and ISUP 4 cancer). Penzkofer et al. suggested a high patient-level sensitivity and NPV (≥90%) as a quality standard for computer-aided detection of csPCa [7]. It is crucial for the algorithm to achieve both high sensitivity and NPV in order to avoid an increase in benign or clinically insignificant biopsies, especially operating at a threshold of PI-RADS ≥ 3. The algorithm employed in our study outperformed in sensitivity (0.97%) and almost met the NPV cut-off (0.88%) for a threshold of PI-RADS > 2, suggesting an adequate and reliable performance. This suggests that the implementation of AI in addition to human reading will not produce surplus biopsies and is therefore more likely to be accepted as a supportive tool instead of being regarded as hindering and time-consuming. Integration in the clinical workflow could increase reader confidence and reduce doubt in equivocal cases, therefore rendering the process more time-efficient. Especially inexperienced readers could benefit from a structured reading workflow in which AI helps to detect, classify, and report in a standardized setting. As soon as supportive AI algorithms are further refined and validated, they could be implemented in the clinical workflow prior to human reading to exclude PCa.

AI measured significantly smaller gland volumes compared to human readers, hence higher PSA densities were derived. Recent other studies suggest at least similar performance of AI volume measurements [24]. The difference to our results might stem from the fact that volumes of the gland measured by human readers are usually derived as an approximation from the ellipsoid formula, whereas in our study the volume was acquired from an exact segmentation of the gland in the biopsy preparation process.

Target volumes, on the other hand, tended to be measured higher by AI, although the differences in both groups were not statistically significant (*p* = 0.46). This may be due to the fact that the software included areas in immediate proximity to the tumor in some cases, although we could not define a systematic error in measurements in this study. The error might also stem from the configuration of the algorithm to perform at high sensitivity at detection and to include more suspicious tissue rather than to miss relevant parts of the tumor.

A major limitation of our study is the relatively small number of included patients. Moreover, histopathological correlation was derived from targeted and systematic biopsies that are currently regarded as the gold standard in PCa detection, but theoretically involve the risk of missing carcinoma that could have been detected in radical prostatectomy, although studies could already prove adequate sensitivity of MRI for the detection and localization of ISUP grade > 2 cancers, particularly in cancers with diameters greater than 10 mm [25,26]. Bratan et al. could show that MRI was less sensitive in the detection of ISUP grade 1 PCa, as less than 30% of ISUP grade 1 cancers under 0.5 mL were discovered in RP [27]. As lesions in our study measured 15 mm in their largest diameter on average and we concentrated on csPCa, we assume our histological approach as suitable regarding the research question. Furthermore, our study was of retrospective character and consisted of a subset of consecutive patients in order to test AI performance in each PI-RADS category. This also implies that the study does not include lesions that were not administered to biopsy due to their inconspicuous appearance in MRI, and that additional lesions detected by the algorithm could not be further verified. Hence, diagnostic performance needs to be interpreted as relative to the study cohort.

## 5. Conclusions

AI-augmented lesion detection and scoring proved to be a robust tool with sensitivity comparable to the radiologists and even outperforming human reader specificity in some cases. In anticipation of refinements of the algorithm and upon further validation, AI-detection could be implemented in the clinical workflow prior to human reading in order to exclude PCa, thereby drastically improving reading efficiency.

## Figures and Tables

**Figure 1 cancers-17-00815-f001:**
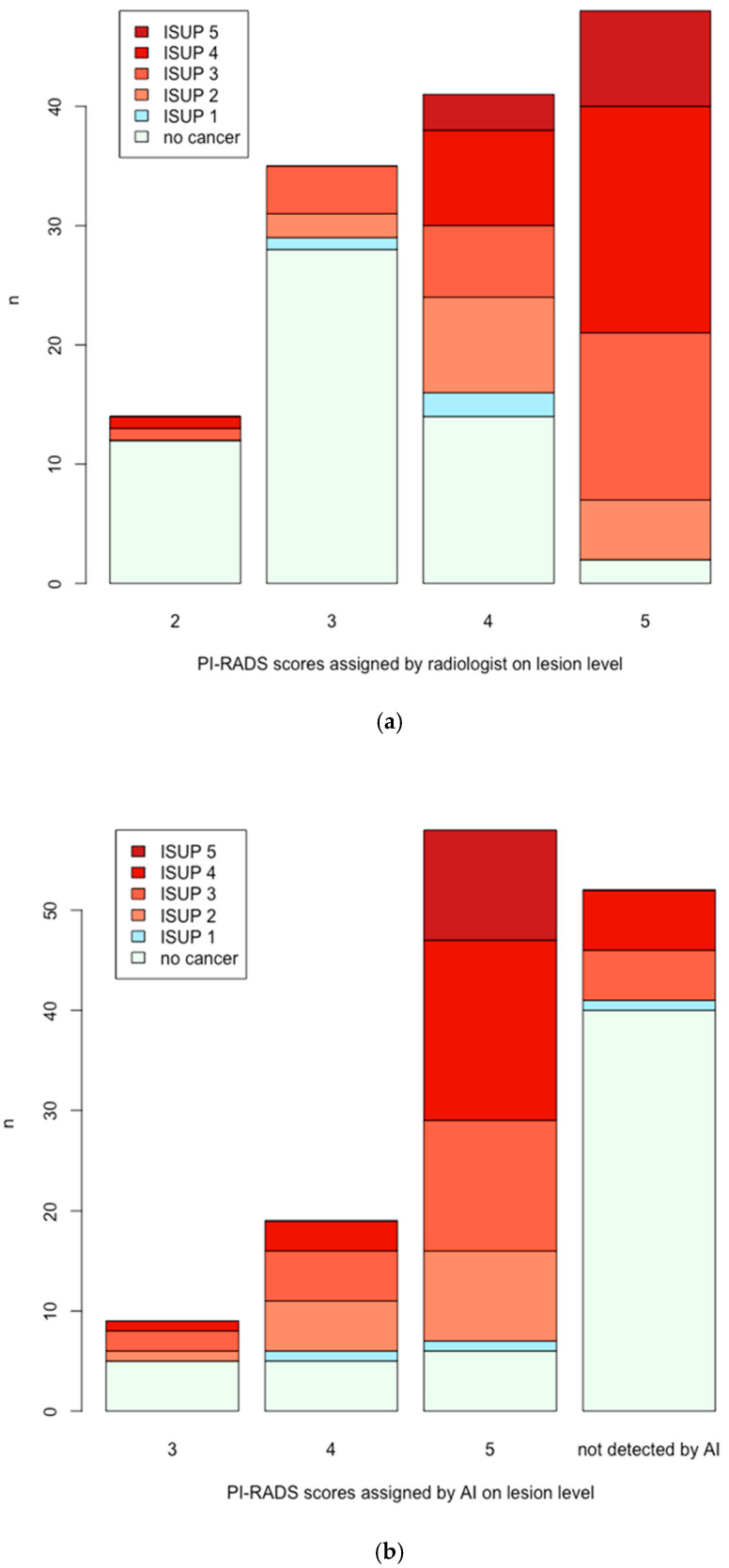
Lesion detection of the radiologist compared to AI. (**a**) Lesions as detected by the radiologists. X-axis: PI-RADS scores as assigned by the reader. Y-axis: number of lesions. (**b**) Lesions as detected by the AI. X-axis: PI-RADS scores as assigned by the AI. Y-axis: number of lesions.

**Figure 2 cancers-17-00815-f002:**
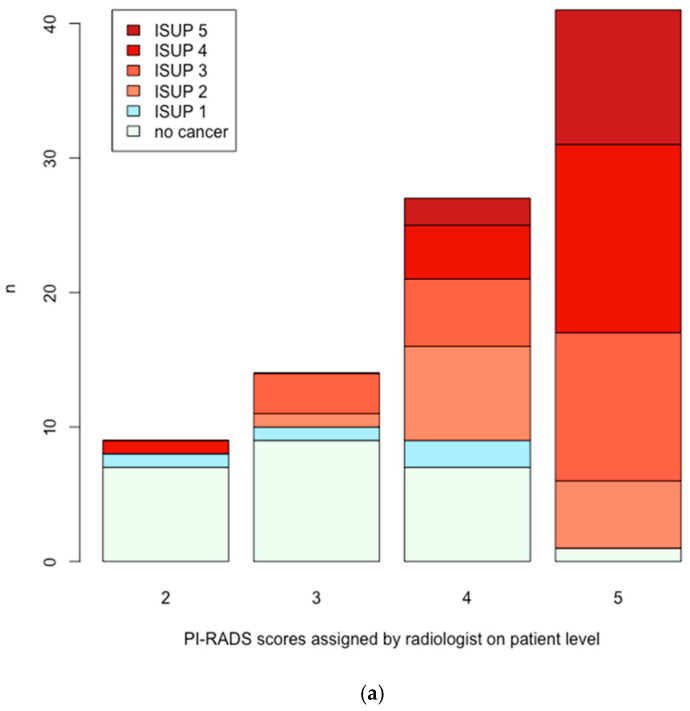
Detection of radiologist compared to AI on patient level. (**a**) PCa as detected by the radiologists. X-axis: PI-RADS scores as assigned by the reader. Y-axis: number of PCa. (**b**) PCa as detected by the AI. X-axis: PI-RADS scores as assigned by the AI. Y-axis: number of PCa.

**Table 1 cancers-17-00815-t001:** Characteristics of the study cohort.

Center		Total	TUE	ULM	FR
*n* patients		91	30	31	30
age (years; median, range)		67	68	69	66
		49–82	49–79	53–82	53–83
PSA at MRI (ng/mL; median, range)		8.4	7.9	9.6	9.2
		1.47–73.7	1.5–44	4.4–73.7	3–52.4
n lesions		138	35	60	43
n lesions in TZ		51	7	30	14
n lesions in PZ		86	28	29	29
n lesions in PZ/TZ		1	0	1	0
n ISUP categories	0	56			
	1	3			
	2	15			
	3	25			
	4	28			
	5	11			
gland volume radiologist (mL; median, range)		52	*p* < 0.001		
		18.45–223			
gland volume AI (mL; median, range)		45			
		16.38–213.82			
PSA density radiologist (ng/mL^2^; median, range)		0.14	*p* < 0.001		
		0.03–1.6			
PSA density AI (ng/mL^2^; median, range)		0.15			
		0.04–1.71			
target volume radiologist (mL; median, range)		0.77	*p* = 0.46		
		0.06–38.22			
target volume AI (mL; median, range)		1.13			
		0.12–36.34			

TUE: Tübingen, ULM: Ulm; FR: Freiburg.

**Table 2 cancers-17-00815-t002:** Comparison of diagnostic accuracies of AI-based and human readings.

Patient Level												
	ISUP	not detected	PI-RADS									
			2	3	4	5	Cut-off	PI-RADS > 2	95% CI	PI-RADS > 3	95% CI	
AI	0	14	0	1	4	5	Sensitivity	0.97	0.89; 1.0	0.91	0.8; 0.96	
	1	1	0	0	2	1	Specificity	0.54	0.34; 0.72	0.57	0.37; 0.76	
	2	0	0	1	3	9	PPV	0.82	0.72; 0.9	0.83	0.72; 0.91	
	3	1	0	2	4	12	NPV	0.88	0.64; 0.99	0.73	0.5; 0.89	
	4	1	0	1	3	14	Accuracy (CI)	0.84	0.74; 0.91	0.8	0.71; 0.88	
	5	0	0	0	0	12						
CDR			0	0.8	0.63	0.89	Cut-off	PI-RADS > 2	95% CI	PI-RADS > 3	95% CI	
radiologist	0		7	9	7	1	Sensitivity	0.98	0.91; 1.0	0.92	0.82; 0.97	
	1		1	1	2	0	Specificity	0.29	0.13; 0.49	0.64	0.44; 0.81	
	2		0	1	7	5	PPV	0.76	0.65; 0.84	0.85	0.75; 0.93	
	3		0	3	5	11	NPV	0.89	0.52; 1.0	0.78	0.56; 0.93	
	4		1	0	4	14	Accuracy (CI)	0.77	0.67; 0.85	0.84	0.74; 0.90	
	5		0	0	2	10	*p* value		0.24		<0.01	
CDR			0.11	0.29	0.67	0.98						
**Target level**												
	**ISUP**	**not detected**	**PI-RADS**									
			2	3	4	5	Cut-off	PI-RADS > 2	95% CI	PI-RADS > 3	95% CI	
AI	0	40	0	5	5	6	Sensitivity	0.86	0.76; 0.93	0.81	0.71; 0.89	
	1	1	0	0	1	1	Specificity	0.70	0.56; 0.81	0.78	0.65; 0.88	
	2	0	0	1	5	9	PPV	0.79	0.69; 0.87	0.83	0.73; 0.91	
	3	5	0	2	5	13	NPV	0.79	0.65; 0.89	0.75	0.63; 0.86	
	4	6	0	1	3	18	Accuracy (CI)	0.79	0.71; 0.86	0.8	0.72; 0.86	
	5	0	0	0	0	11						
CDR			0	0.44	0.68	0.88	Cut-off	PI-RADS > 2	95% CI	PI-RADS > 3	95% CI	
radiologist	0		12	28	14	2	Sensitivity	0.97	0.91; 1.0	0.90	0.81; 0.96	
	1		0	1	2	0	Specificity	0.20	0.11; 0.33	0.70	0.56; 0.81	
	2		0	2	8	5	PPV	0.62	0.53; 0.71	0.80	0.70; 0.88	
	3		1	4	6	14	NPV	0.86	0.57; 0.98	0.84	0.70; 0.93	
	4		1	0	8	19	Accuracy (CI)	0.65	0.56; 0.73	0.82	0.74; 0.87	
	5		0	0	3	8	*p* value		<0.01		0.13	
CDR			0.14	0.17	0.61	0.96						

CDR: cancer detection rates, CI: confidence interval, NPV: negative predictive value, PPV: positive predictive value.

## Data Availability

The raw data supporting the conclusions of this article will be made available by the authors on request.

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
