# Peer review of "Multi-Center Benchmarking of a Commercially Available Artificial Intelligence Algorithm for Prostate Imaging Reporting and Data System (PI-RADS) Score Assignment and Lesion Detection in Prostate MRI"

_cancers, 2025, doi:10.3390/cancers17050815_

Round 1

Reviewer 1 Report

Comments and Suggestions for Authors

The paper is interesting. Methodology is overall correct. There is only two major issues to be discussed.

1) MpMRI of the peripheral zone and of transitional/central zone are practically two different examinations in regard to reader interpretation and accuracy. Therefore, I suggest to discriminate the cohort according to the zone (periphery, lateral, posterior and anterior, central and or transitional zone) and not only to the PIRADS score. I believe that results may be even more interesting if AI were able to overcome the low accuracy of MpMRI of transitional zone.

2) During the co-registration process an error may occur and smaller lesion may be missed. A stratification among PIRADS 3 and 4 lesion by maximum diameter may help to identify the potential error that could be otherwise attributed to the AI or the human reader

Reviewer 2 Report

Comments and Suggestions for Authors
  1. Study Population and Selection Criteria
    • The authors mention a retrospective selection of patients from three university hospitals over a five-year span. Please provide more detail on how many patients were screened initially and how many were excluded. A clear flow diagram (e.g., following CONSORT or a similar approach) would help illustrate the study cohort formation.
  2. Lesion Matching and Ground Truth
    • The study compares AI-detected lesions with radiologist-detected lesions, requiring ≥25% spatial overlap. It would be helpful to clarify the rationale for choosing the 25% threshold. Was this a standard practice in prior studies, or did you test different thresholds?
    • Because only lesions deemed suspicious by radiologists underwent targeted biopsy, how were the AI-exclusive lesions verified as truly negative or false positives? Are there plans for follow-up or additional biopsies if AI detects lesions not identified by radiologists?
  3. Statistical Analysis and Reporting
    • The manuscript would benefit from reporting 95% confidence intervals (CIs) for sensitivity, specificity, PPV, and NPV—especially given the relatively small sample size. CIs would indicate the precision of your estimates and better contextualize the results.
    • While the study demonstrates favorable performance metrics for the AI algorithm, consider a formal comparative test (e.g., McNemar’s test or a similar methodology) to statistically evaluate differences in diagnostic accuracy between the AI and the human readers.
  4. Thresholds and Clinical Impact
    • You report performance at both PI-RADS ≥3 and PI-RADS ≥4 thresholds. Since PI-RADS 3 often represents an equivocal category, it is crucial to discuss how a high sensitivity (with a modest specificity) at ≥3 might influence real-world clinical workflows (e.g., increasing the number of negative or non-significant biopsies).
    • Please provide more clinical context on how you envision the integration of AI-based scoring. For example, if the algorithm has a high negative predictive value, would you propose it as a “rule-out” tool prior to expert reading?
  5. Discussion of Missed Cases
    • The study indicates that the AI missed two cases of clinically significant cancer at the patient level. It would be instructive to elaborate on those missed lesions and consider whether there were any imaging features (e.g., smaller size, lower grade morphological changes) that might have contributed to the AI’s oversight. This insight can guide future algorithm improvements.
